

# The brain-derived neurotrophic factor Val66Met polymorphism increases segregation of structural correlation networks in healthy adult brains

Issei Ueda[1], Kazuhiro Takemoto[2], Keita Watanabe[1], Koichiro Sugimoto[1], Atsuko Ikenouchi[3], Shingo Kakeda[1], Asuka Katsuki[3], Reiji Yoshimura[3] and Yukunori Korogi[1]

[1] Department of Radiology, University of Occupational and Environmental Health, Kitakyusyu, Japan
[2] Department of Bioscience and Bioinformatics, Kyushu Institute of Technology, Iizuka, Japan
[3] Department of Psychiatry, University of Occupational and Environmental Health, Kitakyusyu, Japan

Corresponding author
Issei Ueda, i-ueda@med.uoeh-u.ac.jp

## ABSTRACT

**Background**. Although structural correlation network (SCN) analysis is an approach to evaluate brain networks, the neurobiological interpretation of SCNs is still problematic. Brain-derived neurotrophic factor (BDNF) is well-established as a representative protein related to neuronal differentiation, maturation, and survival. Since a valine-to-methionine substitution at codon 66 of the BDNF gene (BDNF Val66Met single nucleotide polymorphism (SNP)) is well-known to have effects on brain structure and function, we hypothesized that SCNs are affected by the BDNF Val66Met SNP. To gain insight into SCN analysis, we investigated potential differences between BDNF valine (Val) homozygotes and methionine (Met) carriers in the organization of their SCNs derived from inter-regional cortical thickness correlations.

**Methods**. Forty-nine healthy adult subjects (mean age = 41.1 years old) were divided into two groups according to their genotype (n: Val homozygotes = 16, Met carriers = 33). We obtained regional cortical thickness from their brain T1 weighted images. Based on the inter-regional cortical thickness correlations, we generated SCNs and used graph theoretical measures to assess differences between the two groups in terms of network integration, segregation, and modularity.

**Results**. The average local efficiency, a measure of network segregation, of BDNF Met carriers' network was significantly higher than that of the Val homozygotes' (permutation $p$-value = 0.002). Average shortest path lengths (a measure of integration), average local clustering coefficient (another measure of network segregation), small-worldness (a balance between integration and segregation), and modularity (a representative measure for modular architecture) were not significantly different between group (permutation $p$-values $\geq$ 0.01).

**Discussion and Conclusion**. Our results suggest that the BDNF Val66Met polymorphism may potentially influence the pattern of brain regional morphometric (cortical thickness) correlations. Comparing networks derived from inter-regional cortical thickness correlations, Met carrier SCNs have denser connections with neighbors and are more distant from random networks than Val homozygote networks. Thus, it may be necessary to consider potential effects of BDNF gene mutations in SCN analyses.

This is the first study to demonstrate a difference between Val homozygotes and Met carriers in brain SCNs.

# INTRODUCTION

The brain is a complex organ segregated functionally into local areas that differ in their anatomy, and these areas are highly integrated during perception and behavior (*Tononi, Sporns & Edelman, 1994*; *Bullmore & Sporns, 2009*; *Fornito, Zalesky & Breakspear, 2015*). Since the mid-2000s, researchers have conducted analyses from the perspective of anatomical or functional connectivity to understand the neural basis of normal or abnormal brain function (*Sporns, Tononi & Kötter, 2005*; *Bullmore & Sporns, 2009*; *Bullmore & Sporns, 2012*). The current representative methods for analyses of functional networks include correlations of regional functional magnetic resonance imaging (fMRI) or electroencephalography/magnetoencephalography signals (*Evans, 2013*), To analyses of anatomical connectivity, structural network analyses based on a streamline network derived from diffusion-weighted imaging (DWI) and a structural correlation network (SCN) derived from structural magnetic resonance imaging (MRI) are typically adopted (*Evans, 2013*). Research using these connectivity analysis methods is gradually revealing insight into the neural basis of human brain functions and neurological/psychopathological diseases. Such breakthroughs have not been possible with traditional approaches that focused on discrete brain regions (*Fornito, Zalesky & Breakspear, 2015*).

SCN analysis is an approach to evaluate brain networks in vivo. The basic idea of SCN analysis is simple: a single morphological feature, such as cortical thickness or gray matter volume, is measured in each region in multiple subject images, and the correlation of estimates for that feature between each brain region is calculated for each pair of all possible regions (*Evans, 2013*). Analysis of brain networks based on structural correlation is expected to be more reproducible than network analyses based on fMRI or DWI because structural correlation analysis reconstructs the entire-brain network using simpler measurements and calculations. SCN analyses have previously been used to understand brain function (*Tuladhar et al., 2015*; *Chen et al., 2008*), the basis of neuropsychiatric disorders (*Bernhardt et al., 2008*; *Bassett et al., 2008*; *Balardin et al., 2015*; *He, Chen & Evans, 2008*; *Mueller et al., 2015*), and characteristics of brain development (*Khundrakpam et al., 2013*; *Váša et al., 2018*; *Raznahan et al., 2011*). However, neurobiological interpretation of SCNs is still problematic. The basis of SCNs is presumed to involve similarities in terms of cell structure, gene expression, and function, as well as neuronal connectivity (*Alexander-Bloch, Giedd & Bullmore, 2013*; *Evans, 2013*), but the source of this connectivity is unclear. Therefore, it is important to gain further knowledge regarding SCN analyses given that it is one of the more promising approaches to understand the brain.

Brain areas are highly connected through nerve fibers (axons) of nerve cells (neurons). Brain-derived neurotrophic factor (BDNF) is well-established as a representative protein related to neuronal differentiation, maturation, and survival (*Acheson et al., 1995*; *Huang & Reichardt, 2001*; *Binder & Scharfman, 2004*). The human BDNF gene is located on chromosome 11, region p13-14 and it spans ~70 kb. The gene has a complex structure consisting of 11 exons in the 5′ end and nine functional promoters (*Cattaneo et al., 2016*). The human BDNF gene frequently carries a no conserved single nucleotide polymorphism (SNP) that results in a valine-to-methionine substitution at codon 66 (Val66Met). This SNP does not affect BDNF signaling but is reported to cause a deficit in cellular distribution and dysregulated BDNF secretion in neurons (*Egan et al., 2003*; *Kuczewski, Porcher & Gaiarsa, 2010*).

Decreases in activity-dependent release of BDNF are related to the structure and function of the developing brain (*Bath & Lee, 2006*). Several studies have reported the effects of the BDNF Val66Met SNP on gray or white matter structural alterations (*Pezawas et al., 2004*; *Montag et al., 2009*; *Chiang et al., 2011*; *Tost et al., 2013*; *Huang et al., 2014*), and a number of reports show that the BDNF Val66Met SNP affects brain functions as reviewed in *Toh et al. (2018)*. Additionally, given that it is accompanied by brain neural circuitry changes, the BDNF Val66Met SNP is thought to be related to the psychopathology of several neuropsychiatric disorders including major depression (*Frodl et al., 2007*), schizophrenia (*Ho et al., 2006*), and others (*Harrisberger et al., 2015*). Furthermore, the BDNF protein is related to brain development and maturation because BDNF influences neuronal proliferation, differentiation and survival of neurons, as well as neural morphology and function, synaptic revisions, axonal maintenance, and neuroplasticity throughout life (*Egan et al., 2003*; *Frielingsdorf et al., 2010*; *Jasińska et al., 2017*).

Several recent studies describe the effects of the Val66Met SNP on structural and functional networks in the brain (*Ziegler et al., 2013*; *Park et al., 2017*; *Schweiger et al., 2019*; *Franzmeier et al., 2019*). Also, recent cortical correlation network studies on brain developmental changes in adolescence showed that brain integration, segregation and modularity change as a result of the developmental process of pruning combined with consolidation of surviving connections (*Khundrakpam et al., 2013*; *Váša et al., 2018*). Based on these reports, we hypothesized that the BDNF SNP affects segregation, integration, and modular architectures of SCN.

Thus, the purpose of this study was to gain new insights regarding SCN analysis. Given that the BDNF Val66Met SNP is known to affect brain structure and function, we hypothesized that this SNP also affects brain SCNs. We examined whether there are differences between the brains of BDNF methionine (Met) carriers and valine (Val) homozygotes in terms of SCNs.

## MATERIALS AND METHODS

To investigate the effects of BDNF gene mutation on brain SCNs, we created a Val homozygote structural network and a Met carrier network based on inter-regional cortical thickness correlations, and we conducted statistical comparisons of graph theoretical

**Table 1  Genotype frequency, gender, age, and years of education of the two groups.**

| Characteristics | Genotype | |
| --- | --- | --- |
| | Val homozygotes (*n* = 16) | Met carriers (*n* = 33) |
| Gender (M:F) | 13:3 | 23:10 |
| Age [years] (Mean ± SD) | 40.5 ± 11.3 | 41.4 ± 11.5 |
| Years of education (Mean ± SD) | 16.4 ± 3.0 | 16.5 ± 2.3 |

Notes.
  Val, valine; Met, methionine; M, male; F, female; SD, standard deviation.

measures using permutation simulations. See Fig. 1 for an overview of the analysis in this study.

## Participants

The Local Ethics Committee of the University of Occupational and Environmental Health, Kitakyushu, Japan (UOEH) granted Ethical approval to carry out the study (Ethical Application Ref: 第セ H25-13号) A copy of the IRB approval documentation and the blank consent form are provided as supplemental files. All participants were informed about the purpose of the study, and written informed consents were obtained from all subjects via the forms.

Forty-nine subjects of Japanese origin were included in the study (mean age 41.1 years, standard deviation = 11.3 years; range 20–65 years; male:female = 36:13; see Table 1).

Most of the participants were recruited from the psychiatric or radiology departments of UOEH. None of the participants reported any neurological and/or psychopathological disorder (e.g., depression, attention deficit/hyperactivity disorder, and/or schizophrenia) in a simple questionnaire enquiring about their lifetime history of such diseases. All of the participants were right handed. We also asked for the participants' number of years of education; this variable might help to better compare the results of future studies investigating the effect of the BDNF Val66Met polymorphism on the structure of the brain. However, as nearly all participants were medical doctors, no differences in education were observed. Because of the recruitment procedure used (medical doctors, characterized by a higher percentage of male doctors), males were over-represented in the investigated sample. All participants were of East Asian ethnicity.

After undergoing MRI scans, two radiologists with expertise in neuroradiology evaluated the images and confirmed that there were no abnormalities in the participants' brain. Then each participant provided a blood sample for genotyping the BDNF Val66Met polymorphism.

## Genetic analysis

DNA was extracted from the blood samples of each of the 49 participants according to standard laboratory protocols. DNA was isolated from peripheral blood mononuclear cells using the QIAamp (R) DNA Mini-Kit (QIAGEN, Tokyo, Japan). Genotyping

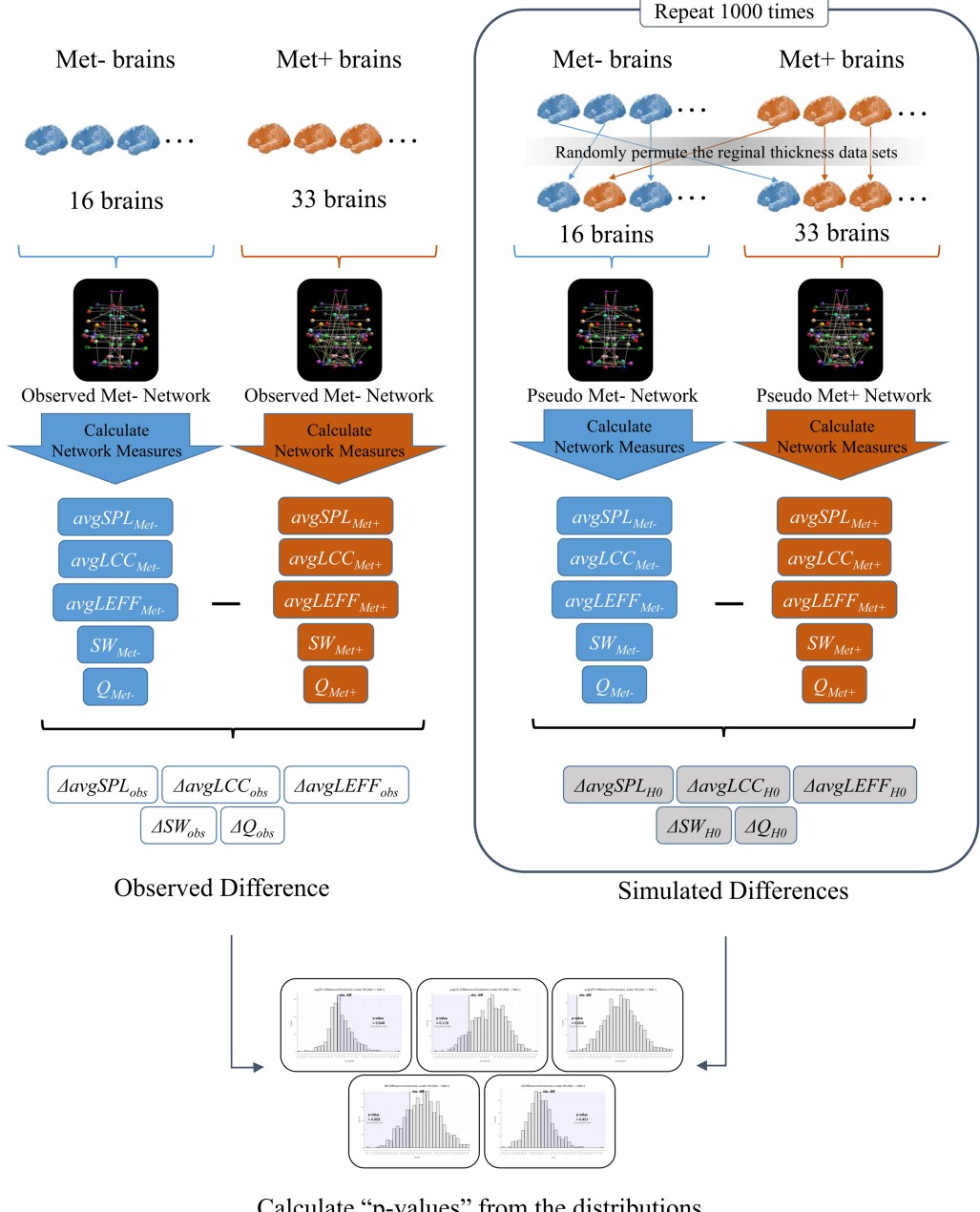

Figure 1 **An overview of the analysis.** First, we created a structural correlation network (SCN) using the FreeSurfer cortical thickness data of 68 brain regions from the BDNF Val homozygotes (Met-) (Observed Met- Network). We also created a SCN of BDNF Met carriers (Met+) using the data from all of the Met carrier subjects in the same manner (Observed Met+ Network). Then, we calculated the average shortest path length (*avgSPL*), average local clustering coefficient (*avgLCC*), average local efficient (*avgLEFF*), small-worldness (*SW*), and modularity (*Q*) of each network, and differences of each network measure between the two networks (continued on next page...)

**Figure 1 (…continued)**
($\Delta avgSPL_{obs}$, $\Delta avgLCC_{obs}$, $\Delta avgLEFF_{obs}$, $\Delta SW_{obs}$ and $\Delta Q_{obs}$). Next, we randomly permuted the cortical thickness data sets from the subjects and partitioned the 49 subjects into two groups. We created a surrogate SCN from the 16 subjects of the first group (Pseudo Met- Network), and another surrogate SCN from the 33 subjects of the second group (Pseudo Met+ Network). We calculated $avgSPL$, $avgLCC$, $avgLEFF$, $SW$, and $Q$ of each pseudo network and subtracted them to get each difference between the two groups ($\Delta avgSPL_{H0}$, $\Delta avgLCC_{H0}$, $\Delta avgLEFF_{H0}$, $\Delta SW_{H0}$, and $\Delta Q_{H0}$). We repeated this permutation simulation 1,000 times and obtained the distributions of the differences of each measure under the null hypothesis (i.e., the hypothesis that the correlation pattern of each region is the same between the two groups). Finally, we examined significance of the observed difference of each measure by the one-tailed $p$-values calculated as the rate of entries that were greater than (or smaller than) the observed between-group difference in the 1,000 times permutations.

was carried out with a polymerase chain reaction single nucleotide polymorphism (SNP) genotyping system using a BigDye Terminator v3.1 Cycle Sequencing Kit (Life Technologies Japan, Tokyo, Japan). The DNA was read using a BMG Applied Biosystem 3730xI DNA Analyzer (Life Technologies Japan, Tokyo, Japan). We used a forward primer (ATGAAGGCTGCCCCCATGAAA) and a reverse primer (TGACTACTGAGCATCACCCTG) for the BDNF Val66Met polymorphism. The participants were either homozygous for the Val allele (Val/Val genotype), heterozygous (Val/Met genotype), or homozygous for the Met allele (Met/Met genotype).

The 49 study subjects were divided into groups based on their BDNF genotype. Because the Met/Met genotype is less prevalent in Japan (about 15.9% (*Shimizu, Hashimoto & Iyo, 2004*)), participants were grouped according to the occurrence of the Met allele, which resulted in two independent groups: Val homozygotes (Val/Val genotype) and Met carriers (Val/Met and Met/Met genotypes).

## Analyses of demographic and genotyping data

To assess the demographics and characteristics of participants, a nonparametric rank test (Wilcoxon–Mann–Whitney test) was applied to compare the differences in age and years of education among the subjects that were either Met carriers or Val homozygotes. Fisher's exact test was used to evaluate the difference in gender among the groups. To assess the existence of any sampling bias within our subjects, we confirmed that the genotype distributions did not deviate from the expected frequency under Hardy–Weinberg equilibrium and tested the expected Japanese genotype/allele frequencies (*Shimizu, Hashimoto & Iyo, 2004*) of the BDNF gene polymorphism using chi-square tests.

These statistical analyses were performed using R software (version 3.4.1, R Foundation for Statistical Computing, Vienna, Austria). The level of significance was set to $p < 0.05$ for all demographic experiments.

## MRI acquisition

MRI scans were performed on a 3.0-Tesla scanner (Signa EXCITE 3T; GE Healthcare, Milwaukee, WI, USA) using a dedicated eight-channel phased-array coil (USA Instruments Aurora, OH, USA). Three-dimensional fast spoiled gradient recalled acquisition with steady state (3D-FSPGR) images of the whole head were obtained with the following
parameters:10/4.1/700 (repetition time ms/echo time ms/inversion time), a flip angle of 10, a 24- cm field of view and 1.2-mm-thick sections with $0.47 \times 0.47 \times 0.6$ mm$^3$ resolution. In addition, diffusion tensor images (DTIs) were also obtained assuming other hypothesis study. Although these DTIs were not used for this study's analysis, the images were used to check whether the subject brains had any abnormalities.

### Regional cortical thickness measurement

We used the FreeSurfer software package (version 6.0, https://surfer.nmr.mgh.harvard. edu/) for cortical surface reconstruction and cortical thickness estimation. Because detailed procedures for using FreeSurfer have previously been by authors such as (*Dale, Fischl & Sereno, 1999*; *Fischl & Dale, 2000*; *Desikan et al., 2006*; *Balardin et al., 2015*), we describe them here only briefly. The FreeSurfer processing stream consists of the co-registration of the subject T1 image to an atlas for bias correction and intensity normalization, and for brain extraction/removal of non-brain tissue. The cortical boundaries between the gray and white matter, and between the gray matter and cerebrospinal fluid, were tracked, tessellated and smoothed to produce a surface mesh. Topology correction and surface deformation were applied and cortical thickness was estimated by considering the closest distance from the gray/white matter boundary to the gray matter/cerebrospinal fluid boundary at each vertex on the tessellated surface. The cortical surface of each hemisphere was then parcellated based on gyral and sulcal landmarks according to the Desikan-Killiany atlas (*Desikan et al., 2006*). After FreeSurfer processing, 34 separated cerebral regions for each hemisphere (a total of 68 regions) were identified. The 68 brain regions' names and the abbreviations for these names are summarized in Table S1.

Because a research member who was an expert in using FreeSurfer processing visually checked the final FreeSurfer outputs for accuracy or processing failures in a systematic manner, and all of the subjects' outputs were confirmed to be correctly labeled and measured. Thus, we obtained 49 regional thickness values for each of the 68 brain regions.

Prior to the construction of structural correlation networks, FreeSurfer cortical thicknesses were corrected using the following steps: first, in order to remove the effects of age, gender, and an age-gender interaction, linear regression model analyses was conducted at every region with the best model based on the Akaike information criterion (AIC). Next, the residuals of the linear models were substituted for the raw cortical thickness values.

We examined the statistical differences in the cortical thickness of each region between the two groups in a Analysis S1. The results are summarized in Table S2.

### Network construction

For network construction, we first defined anatomical connections as statistical associations in cortical thickness between the brain regions in order to characterize human brain networks. Such a morphometry-based connection concept has been described in many previous studies (*Worsley et al., 2005*; *Lerch et al., 2006*; *He, Chen & Evans, 2007*; *Váša et al., 2018*).

We evaluated statistical similarities in cortical thickness between all region pairs using Spearman correlation coefficients ($r$) across subjects in each group. The correlation

coefficient $r$ ranges in value from 1 (positive correlation) to $-1$ (negative correlation), and the closer to 1 or $-1$ the more significant the correlation. However, to avoid the complication of statistical feature descriptions in subsequent graph theoretical analyses, we converted $r$ values into absolute values. (As a supplementary analysis, we conducted a negative-edge-removed version analysis to check if the results change when edges with negative weight were removed. Please refer to Analysis S1). In addition, we conducted edge selection based on Random Matrix Theory (RMT) as inspired by *Deng et al. (2012)*. RMT was first proposed by Winger and Dyson as a powerful method to identify and model phase transitions associated with disorder and noise in statistical physics and material science (*Wigner, 1967*). The RMT method is able to automatically identify a threshold to construct a mathematically meaningful network from a correlation matrix. We used a RMT-based approach to conduct threshold scanning. Correlation coefficients greater than the threshold were retained, and the remaining coefficients were set to zero.

Thus, two inter-regional correlation matrices ($N \times N$, where $N$ is the number of brain regions, here $N = 68$) consisting of correlations between every pair of brain regions were acquired using the 16 subjects' brains included in the Val homozygote group and 33 subjects' brains included in the Met carrier group respectively.

To perform graph theoretical analyses, we used the anatomical inter-regional correlation matrices obtained above as the adjacency matrices of undirected weighted graphs. Nodes and edges represented brain regions and connections between the regions, respectively. The edges weight $w_{ij}$ (an element of the adjacency matrix) corresponds to $|r|$ between brain regions $i$ and $j$; however, they were set as $1/|r|$ when calculating shortest path lengths.

### Network comparison

For network comparison, we first visualized the comparisons between pairs of networks using heat maps and circular graphs. The heat maps were made with R-package *ggplot2* (version 3.2.1; https://ggplot2.tidyverse.org/). The circular graphs were made using the Python software program (version 3.6.8; http://www.python.org) and the *MNE-Python* (v0.19) package (http://www.mne.tools/stable/index.html) (*Gramfort et al., 2013*).

Next, we compared the networks using network analysis. According to our hypothesis on the BDNF SNP's effects on integration, segregation, and modularity, we examined the differences in average shortest path length (*avgSPL*), average local clustering coefficient (*avgLCC*), average local efficiency (*avgLEFF*), small-worldness (*SW*), and modularity (*Q*) inspired by previous studies (*Latora & Marchiori, 2001*; *Rubinov & Sporns, 2010*; *Rubinov & Sporns, 2011*). *avgSPL* is a representative measure for integration, and *avgLCC* and *avgLEFF* are well used measures of segregation. *SW* is a balance between *avgSPL* and *avgLCC, and Q* is a representative measure for the modular architecture of a network. We computed these network measures using R software version 3.3.1 (https://www.R-project.org) and the R-package *igraph* version 1.0.1 (http://www.igraph.org).

*avgSPL* is the average shortest path length among all reachable node pairs and was calculated using the function shortest.path in the *igraph* package.

*avgLCC* indicates the average strength of local-scale connectivity and is defined as $\frac{1}{N}\sum_{i=1}^{N}C_i^w$, where $C_i^w$ is a weighted version of the nodal clustering coefficient defined

as $C_i^w = \frac{1}{s_i(k_i-1)} \sum_{j,h\in nn(i), i\neq j\neq h} \frac{w_{ij}+w_{ih}}{2} sign(w_{ij} w_{ih} w_{jh})$. Here $nn(i)$ is the set of the nearest neighbors of node $i$. $k_i$ is the degree of node $i$; $k_i = \sum_{j=1}^{N} sign(w_{ij})$, and $sign(x)$ is the sign function. $C_i^w$ was computed using the function *transitivity* in the *igraph* package.

*avgLEFF* indicates average information efficiency among nearest neighbor (i.e., at local level), and in is defined as $\frac{1}{N} \cdot \sum_{i\in N} E_{loc}^w(i)$, where $E_{loc}^w(i)$ is the local efficiency of node $i$ defined as $E_{loc}^w(i) = \frac{\sum_{j,h\in nn(i), j\neq h}\left[d_{jh}^w\right]^{-1}}{k_i(k_i-1)}$, where $d_{ij}^w$ is the shortest path length between $i$ and $j$ in a weighted network.

*SW* (*Humphries, Gurney & Prescott, 2006*) was proposed as a measure of the small-world property in real-world networked systems (*Watts & Strogatz, 1998*). The small-world property indicates that all node pairs in a network were reachable by a short distance (i.e., the distance expected from random networks), even though the network is divided into highly interconnected clusters (i.e., the network is far from a random network). Specifically, SW was calculated based on the *avgSPL* and *avgLCC* in the actual and randomized networks: $(avgLCC_{act}/avgLCC_{rand})/(avgSPL_{act}/avgSPL_{rand})$, where $X_{act}$ represents a network measure $X$ (i.e., *avgLCC* or *avgSPL*) in the actual networks, and $X_{rand}$ is the average $X$ obtained from 100 randomized networks. For an actual network, the randomized networks were generated by permutation of the edge weights in the original network.

The modularity of networks was measured using the $Q$-value, an excellent guide to whether a particular division of a network into communities is strong or weak division (*Newman, 2004a*; *Newman, 2004b*). We considered a weighted version of the $Q$-value (*Fortunato, 2010*). The $Q$ value is defined as the fraction of edge weights that lie within, rather than between, modules relative to that expected by chance as follows:

$$Q = \frac{1}{2W} \sum_{ij} \left(w_{ij} - \frac{s_i s_j}{2W}\right) \delta(c_i, c_j),$$

where $W$ is the sum of the weights of all edges, and $\delta(c_i, c_j) = 1$ if nodes $i$ and $j$ belong to the same module but 0 otherwise. A network with a higher $Q$ indicates a higher modular structure. In the present study, an algorithm based on simulated annealing (*Guimerà & Nunes Amaral, 2005*) was used to find the maximum $Q$ in order to avoid the resolution limit problem in community (or module) detection (*Fortunato & Barthélemy, 2007*; *Fortunato, 2010*) wherever possible. The maximum $Q$ was defined as the network modularity of brain networks. We used the *netcarto* function in the R-package *rnetcarto* (version 0.2.4) to compute network modularity $Q$.

We did not consider regional network characteristics (i.e. local network measures) because we could not find any report on the BDNF SNP's effects on specific brain regions. To show statistically significant differences in the network measures between Val homozygotes and Met carriers, we calculated type I error rates using the permutation method under the null hypothesis. Details of the statistical method are described in the next section.

## Network measure comparison with permutation method

To statistically test the difference in the network measures between the Met carriers' network and the Val homozygotes' network, we considered a permutation test approach

for weighted correlation networks (*Bassett et al., 2008*; *Gill, Datta & Datta, 2010*). This approach randomly permutes the regional cortical thickness data sets from the subjects, and contrasts two networks built by partitioning the 49 subjects into two groups. We calculated the network measures from the permuted networks and obtained the approximate *p*-value by computing the probability that the simulated differences are greater than or equal to the observed difference. That is,

$$p\left(DiffNM_{simulated(i)} > DiffNM_{observed}\right) = \frac{1}{tpc}\sum_{i=1}^{tpc}I\left(DiffNM_{simulated(i)} > DiffNM_{observed}\right)$$

$$p\left(DiffNM_{simulated(i)} < DiffNM_{observed}\right) = \frac{1}{tpc}\sum_{i=1}^{tpc}I\left(DiffNM_{simulated(i)} < DiffNM_{observed}\right)$$

where $p(x)$ is the *p*-value under the condition *x*. *tpc* is the total permutation count (in this study, *tpc=1000*). $DiffNM_{simulated(i)}$ is the difference between the network measures calculated from the permutated Met carriers' network and permutated Val homozygotes' network obtained at *i* th trial, and $DiffNM_{observed}$ is the difference in the network measure between the observed Val homozygote and Met carrier networks. The *p*-values of every network measure were computed simultaneously using the same set of random permutations.

Under the null hypothesis the correlation pattern of each region is the same: the hypothesis was tested based on this permutation scheme being confirmed. That is, between the pseudo-two groups generated by permutation simulation, we calculated the network measure differences, and we examine the frequency that each simulated difference is greater (or less) than the observed difference. If the frequency is less than the predetermined significant level, then we reject the null hypothesis that there is no significant difference between the two groups, and we accept the alternative hypothesis that there is a significant difference between them. A one-sided nonparametric permutation test was used to test for global differences of the network measures between the two groups as specified by the hypotheses. The number of the repetition time for the permutation tests was set to 1,000 based on the rules of thumb and the load on the computer. We controlled the familywise error rate with the Bonferroni correction; the critical value (alpha) for an individual test by dividing the familywise error rate (0.05) by the number of tests. Since we conducted five statistical tests respecting global network measures in this study, the critical value for an individual test was 0.05/5 = 0.01, and we only consider individual tests with *p*-value < 0.01 to be significant. Statistical analyses were performed by using R software (version 3.6.1, R Foundation for Statistical Computing, Vienna, Austria).

## RESULTS

### Demographic and genotyping data

Table 1 displays the genotype frequency, gender, age, and years of education in each of the two groups. Age, years of education, and gender were not significantly different in the two genotype groups ($p$-values = 0.73, 0.996, and 0.508 respectively).

Furthermore, the genotype distribution of the Val66Met polymorphism in our healthy Japanese cohort comprised 16 Val/Val homozygotes (32.7%), 28 Val/Met heterozygotes (57.1%), and 5 Met/Met homozygous subjects (10.2%). The distribution of genotypes did not differ from the frequencies expected from Hardy-Weinberg equilibrium ($\chi^2 = 2.03$, df = 1, $p$-value = 0.363). The allele and genotype frequencies are consistent with those reported previously for Japanese individuals (*Shimizu, Hashimoto & Iyo, 2004*) (allele: $\chi^2 = 0.109$, df = 1, $p$-value = 0.741; genotype: $\chi^2 = 1.478$, df = 2, $p$-value = 0.478).

### Network comparison

First, to understand the overviews of the structural correlation networks of both groups, we constructed heat maps (Fig. 2). When the heat maps of the two groups were compared, a grid pattern was more clearly perceived in the Met carrier heat map than in the Val homozygote heat map. In a heat map of the brain network, the grids represent sub-divided brain regions (i.e., modules). From these grid patterns, we observed that the Met carrier brain network had clearer module architecture than the Val homozygote brain network. When we focused on the colors in each heat map, the Met carriers' map had more red areas whereas the Val homozygote heat map had more green areas. This observation suggested that the Val homozygote networks consisted of middle connections whereas the Met carrier networks had more connections with high strength.

Next, we constructed circular graphs using the 100 strongest connections (Fig. 3). In both groups, each brain region had a strong correlation with the counterpart of the region which is represented by the white lines straddling the midline of the each circular graph. This finding seemed to reflect the anatomical left-right symmetry of the cerebrum. In addition, the Met carrier network had fewer brain regions related to the 100 strongest connections than the Val homozygote network. When we focused on the color of the edges, there is no apparent difference between the Val homozygote graph and Met carrier graph.

The results of the statistical comparisons of network measures related to integration, segregation (i.e., *avgSPL*, *avgLCC*, *avgLEFF*, and *SW*), and modular architecture (i.e. *Q*) are given in Table 2. The *avgLEFF* of the Met carriers' network was larger than that of the Val homozygotes' network, and the statistical difference between the *avgLEFF* of the two networks was significant ($p$-value $< 0.01$). Although the Met carriers network's *avgLCC* and *SW* were larger, and *avgSPL* and *Q* were smaller, than those of the Val homozygote network, there were no significant differences between the two groups in regard to these measures ($p$-value $\geqq 0.01$). The permutation distributions and permutation $p$-values for these network measures are shown in Fig. 4.
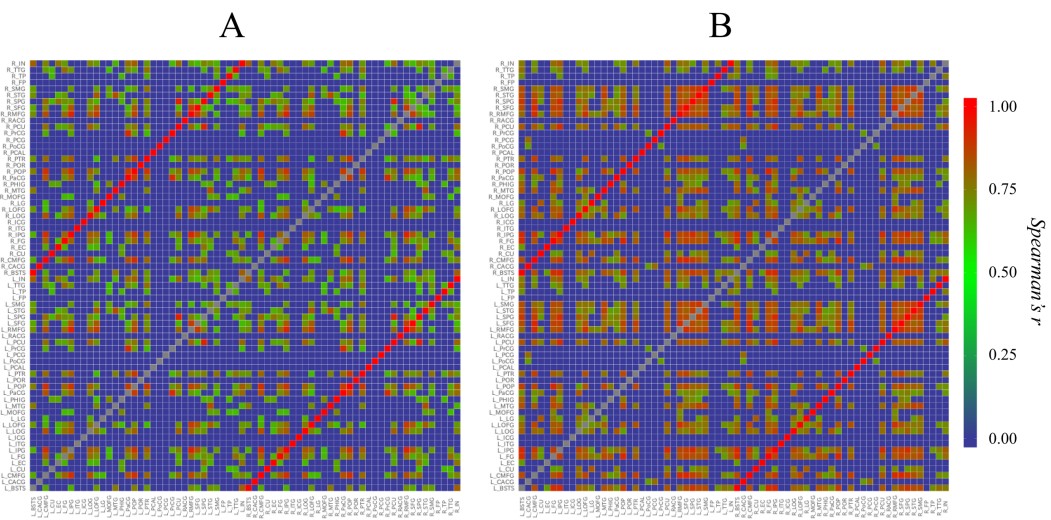

**Figure 2** **Heat maps of the brain structural correlation network of the BDNF Val homozygotes (A) and Met carriers (B).** These heat maps represent each strength of inter-regional cortical thickness correlation of 68 brain regions. The color scale corresponds to the absolute values of Spearman's correlation coefficient. In a heat map of the brain network, grids represent sub-divided brain regions (i.e., modules). From these grid patterns, we observed that the Met carrier brain network had clearer module architecture than the Val homozygote brain network. When we focused on colors in each heat map, the Met carriers' map had more red areas whereas the Val homozygote heat map had more green areas. This observation suggested that the Val homozygote networks consisted of middle connections whereas the Met carrier networks had more connections with high strength. In both groups, each brain region had the strong correlation with the counterpart of the region; these correlations are described in the heat maps as the red lines running diagonally. The abbreviations of the brain regions used in these heat maps are summarized in Table S1.

## DISCUSSION

We examined whether the BDNF Val66Met SNP makes any difference to the brain structural network, and we found that this SNP influences the correlation patterns of the regional cortical thickness; the *avgLEFF* (a measure of network segregation) of the BDNF Met carriers' network was significantly higher than that of the Val homozygotes'. *avgLEFF* is the average of nodal local efficiency of all nodes in a network. The nodal local efficiency is a measure of segregation based on calculated from the average inverse shortest path length to all neighbor nodes (*Rubinov & Sporns, 2010*; *Latora & Marchiori, 2001*), and the average of all nodes (i.e., avgLEFF) can be regarded as a measure of segregation of a whole network. Thus a network with higher *avgLEFF* has greater local efficiency and is thought to have less random topology (*Fornito, Zalesky & Breakspear, 2013*). Therefore, our data suggest that the Met carriers' structural correlation networks have denser connections with neighbors and are more distant from random networks than are the Val homozygotes' networks.

In contrast to the *avgLEFF*, we found no significant differences in the avgLCC, *avgSPL*, and *SW* of Met carriers' and Val homozygotes' network. *avgLCC* is the average of nodal clustering coefficients of all nodes in a network, and the nodal clustering coefficient is a measure of segregation based on the number of triangles around an individual node (*Watts*

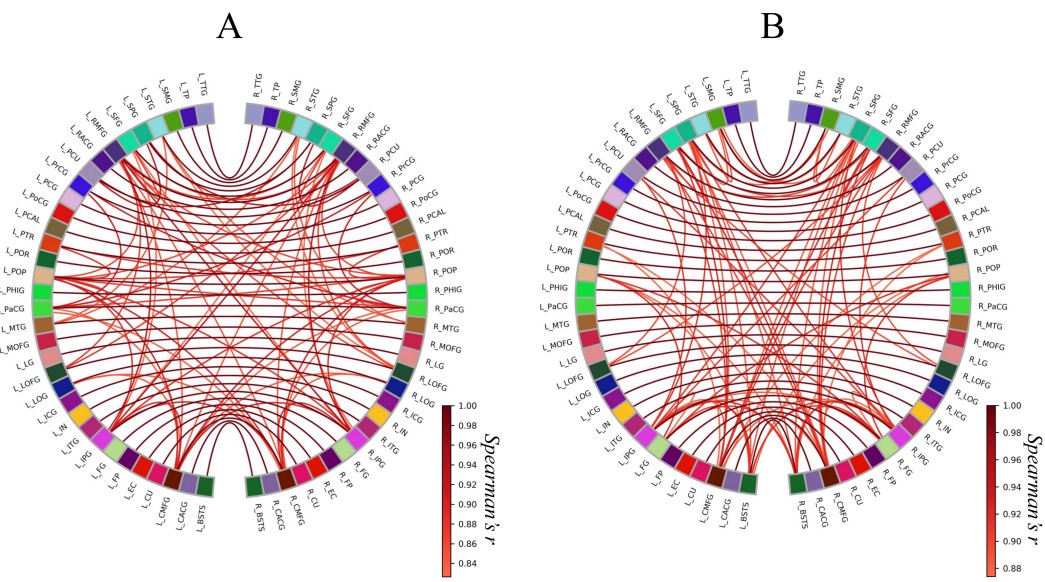

**Figure 3** **Circular graphs of brain structural correlation network of the BDNF Val homozygotes (A) and Met carriers (B).** These circular graphs are describing the inter-regional cortical thickness correlation, where the nodes represent brain regions, and the edges represent undirected connections between the regions. Each of the edge weight corresponds to the absolute value of Spearman's correlation coefficient. In these circular graphs, we only showed the 100 strongest connections. The Met carrier network had fewer brain regions related to the 100 strongest connections than the Val homozygote network. The abbreviations of the brain regions used in these circular graphs are summarized in Table S1.

**Table 2** **The statistical comparison result regarding network measures of the brain structural correlation networks from the BDNF Val homozygotes and Met carriers.**

| Weighted Global Network Measure | Val Homozygotes | Met Carriers | *p*-value | |
|---|---|---|---|---|
| Average Shortest Path Length (*avgSPL*) | 1.814 | 1.657 | 0.549 | |
| Average Local Clustering Coefficient (*avgLCC*) | 0.551 | 0.715 | 0.118 | |
| Average Local Efficiency (*avgLEFF*) | 0.484 | 0.627 | **0.002** | * |
| Small-worldness (*SW*) | 0.997 | 1.409 | 0.252 | |
| Modularity (*Q*) | 0.190 | 0.119 | 0.451 | |

**Notes.**
*The bold text with an asterisk indicates significance (*p*-value < 0.01).
  BDNF, brainderived neurotrophic factor; Val, valine; Met, methionine.

*& Strogatz, 1998*). *avgSPL* is the most commonly used measure of network integration (*Rubinov & Sporns, 2010*); the calculation formula reflects the average distance between all pairs of nodes in the network. *SW* is a network property that quantify the balance between integration and segregation (*Lord et al., 2017*). Our results for these measures suggest that brain structural correlation networks maintain a balance between segregation and integration even when they have the BDNF Val66Met SNP. However, considering the

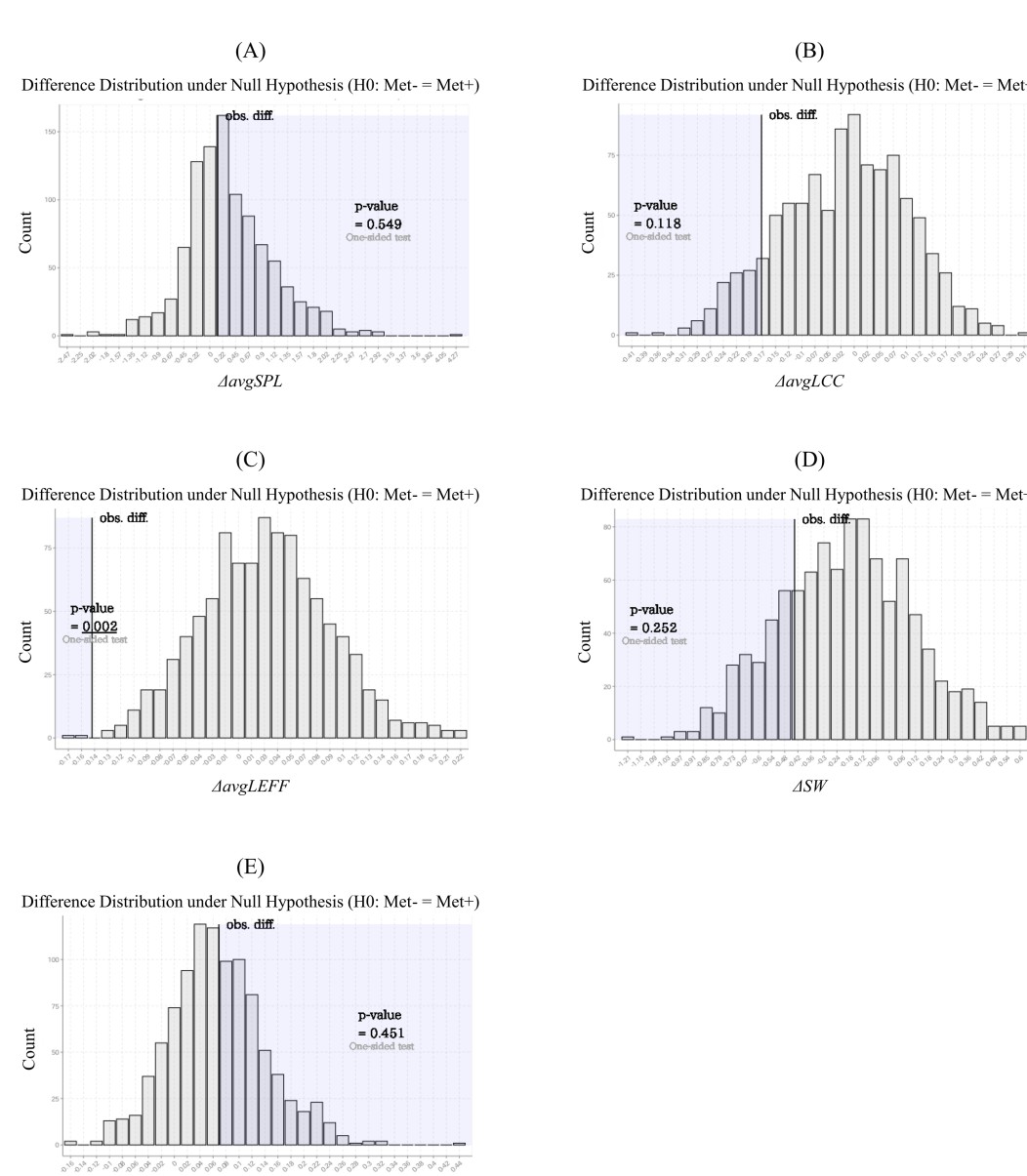

**Figure 4 Permutation distributions and *p*-values of the network measures calculated from the structural correlation networks.** These histograms (A-E) were obtained by repeating the calculation of the network measure difference between the randomly-partitioned groups of subjects. Each of the *p*-values was calculated as the frequency that the simulated differences were greater (or less) than each of the observed differences. The average local efficiency (*avgLEFF*) of the Met carriers' network was larger than that of the Val homozygotes' network, and the statistical difference between the *avgLEFF* between the two networks was significant (*p*-value = 0.002) (C). There were no significant differences between the two groups in regard to the average shortest path length (*avgSPL*), the average local clustering coefficient (*avgLCC*), small-worldness (*SW*), and modularity (*Q*) (*p*-value $\geqq$ 0.01) (A, B, D, and E).

relatively low *p*-values of the *avgLCC* and *SW* differences, there remains some possibility that the degree of the balance between network segregation and integration would differ if Val homozygote and Met Carrier structural correlation networks were compared using a greater sample size.

We also found no significant difference in modularity (*Q*) between the Met carriers' and Val homozygotes' networks. Modularity is a measure used to quantify the degree to which the network may be sub-divided into clearly delineated and non-overlapping groups (*Newman, 2004a*; *Rubinov & Sporns, 2010*). Although a heat map comparison gave the impression that the Met carriers' brains had more clear modular architecture, the difference could not be confirmed statistically. Thus, the implied result from the heat map was not reflected in the modularity value we calculated. Given the modularity formula used in this study, the lack of a difference between the modularity values of the two networks only indicates that the strengths of the intra- and inter-module connectivities are almost equal, but it does not indicate that other modular features (e.g., number, member or size) are equal. Thus, the lack of modularity difference we found does not necessarily disagree with our first impression from the heat maps. To examine potential modular differences in more detail, we would need to introduce other concepts such as pre-defined module analysis, which has been conducted in previous studies (*Park et al., 2017*; *Váša et al., 2018*), or network analysis using the measurement of well-connected within module and well-distributed between module, which was suggested by *Guimerà & Nunes Amaral (2005)*.

This is the first study to demonstrate a difference between Val homozygotes and Met carriers in brain structural correlation network. However, to our knowledge, two DWI studies have previously examined the effects of the BDNF Val66Met SNP on the brain structural network. Using DWI, Ziegler et al. examined differences in structural brain connectivity between Met carriers and Val homozygotes, and they showed that Met carriers' structural connectivity was greatly increased throughout the forebrain (*Ziegler et al., 2013*). In addition, Park et al. assessed the effect of the SNP on the network properties and robustness of the structural networks derived from white matter tractography (*Park et al., 2017*), and they found that the network of the Met carriers group showed higher vulnerability than Val homozygote group to targeted removal of central nodes.

Here, we did not compare our results with those of these previous studies because they relied on binary networks constructed based on diffusion tensor images. Moreover, the network features considered in the previous studies differed from those considered in our study. However, the hypothesis stated in Ziegler's study (*Ziegler et al., 2013*) can be closely related to our results as a plausible explanation for how the BDNF Val66Met SNP causes increased segregation of structural correlation networks. Specifically, Ziegler et al. stated that brain connectivity might eventually be less profoundly shaped by experience in Met carriers than in Val homozygotes if silent axons are relatively less likely to be pruned due to reduced BDNF secretion in the Met carriers. This hypothesis was based on the BDNF Val66Met SNP being known to cause a reduction in activity-dependent BDNF secretion (*Egan et al., 2003*), as well BDNF studies reporting involvement in long-term potentiation and synaptic plasticity (*Patterson et al., 1996*) and axonal pruning and

maintenance triggering the elimination of synaptically silent axonal terminal arbors (*Cao et al., 2007*; *Singh et al., 2008*). The mechanism hypothesized by Ziegler et al. could explain our finding that Met carriers' brains have greater *avgLEFF*. Perhaps the BDNF Val66Met SNP leads to a reduction in pruning and maintenance of neurons, and the regional connectivities in Met carriers' brains are less enhanced by experience than those in Val homozygotes' brains. As a result, Met carriers' brains develop denser connections among neighboring regions and form a highly clustered network compared to that in Val homozygote brains. However, despite this hypothesis, the detailed mechanism that underlies the BDNF Val66met SNP's effects on brain structural networks cannot be derived from our data; further studies are therefore required.

When utilizing a SCN analysis, researchers have to select appropriate parameters according to the individual study. Although the methodological concept of SCN is relatively simple, there are many parameters that may affect the results of the analysis. These parameters include: definitions of nodes and edges, a measure for quantifying correlations, a method for setting thresholds, a presence or absence of binarization processing, focus of network measures, and other examples. These technical aspects of SCN need to be further examined in future studies. In the current study, for example, the absolute value of the correlation was defined as the weight of each edge based on previous studies (*Balardin et al., 2015*; *He, Chen & Evans, 2008*). However, the biological basis of the correlation between two regions is still not clear, and it is difficult to determine a better way to treat the negative correlation. We performed a supplementary analysis (Analysis S1) to confirm that our results were not distorted by treating connections with opposite (i.e., negative) directions as equivalent to connections with the same (i.e., positive) direction. Analysis S1 shows results following pre-processing to remove the negative correlation. These results were similar to the main analysis and suggest that treating negative correlations the same as positive correlations did not affect the conclusions of this study. Regarding network measures, for another example, we only focused on five network measures (*avgSPL*, *avgLCC*, *avgLEFF*, *SW*, and *Q*), but if we had also focused on other measures such as vulnerability or centrality, we may have captured other differences between the two groups.

One limitation of our study is the relatively small sample size. Although all participants had almost similar backgrounds, and we statistically confirmed that there were no differences between the groups in age, gender and years of education, there is a possibility that the difference we found between Val homozygotes and Met carriers derives from another unknown variable. Thus, a larger sample size would be needed to remove potential effects due to age, gender, cognitive function, and other individual characteristics. In addition, our result should be validated through comparisons of inter-regional correlations in terms of cortical cytoarchitectural or gene expression pattern, comparisons between networks derived from DWI or fMRI. The comparison composition for the current study was limited. Following the related previous studies (*Ziegler et al., 2013*; *Park et al., 2017*), we compared SCNs between groups under the composition of Val homozygotes versus Met carriers in this study. However, given reports that the Val66Met SNP affects the regulation of BDNF secretion (*Egan et al., 2003*; *Kuczewski, Porcher & Gaiarsa, 2010*) and that there were differences in brain structures compared between the three groups (*Forde*

*et al., 2014*), it is possible that the three groups (i.e., Val/Val, Val/Met, and Met/Met) have different network features. Ideally, we should have extended the analysis to the three-group comparison, but due to a sample size imbalance between the three groups in our study, we did not perform such an analysis. Furthermore, a SCN can only be used for group analyses because it is created from group data and not individual data. This practical limitation hinders the application of results from SCN analyses to directly analyze individual brain networks. However, SCN analyses do have the advantage of simplicity compared to analyses of networks derived from DWI or fMRI. Moreover, there are several studies using methods with multiple morphometric features for generating SCNs in individual subjects (*Seidlitz et al., 2018*; *Li et al., 2017*). Finally, while it would be beneficial to evaluate the statistical power and effect size, approaches for comparison of SCNs are not currently established, but in the future bootstrap approaches may be useful. However, this limitation poses little problem given that numerical simulations using synthetic data and previous application studies (*Luo et al., 2006*; *Deng et al., 2012*) confirm the validity of the RMT-based method.

## CONCLUSION

Given that the brain is a network, methods involving network analyses are expected to be able to evaluate features that cannot be captured by simply comparing individual regions to each other. SCN analysis is one methods to evaluate the brain as a network. The results of our study suggest that the BDNF Val66Met SNP affects overall segregation in SCNs. Thus, it may be necessary to consider the effect of BDNF gene mutations in future analyses using SCNs. Overall, our results provide further knowledge regarding the analysis of networks derived from brain structural correlations.

### Funding

This work was supported by a Grant-in-Aid for Scientific Research on Innovative Areas (Comprehensive Brain Science Network) from the Ministry of Education, Science, Sports and Culture of Japan. The funders had no role in study design, data collection and analysis, decision to publish, or preparation of the manuscript.

### Grant Disclosures

The following grant information was disclosed by the authors:
Grant-in-Aid for Scientific Research on Innovative Areas (Comprehensive Brain Science Network).
Ministry of Education, Science, Sports and Culture of Japan.

### Competing Interests

The authors declare there are no competing interests.

## Author Contributions

- Issei Ueda conceived and designed the experiments, performed the experiments, analyzed the data, prepared figures and/or tables, authored or reviewed drafts of the paper, and approved the final draft.
- Kazuhiro Takemoto performed the experiments, analyzed the data, prepared figures and/or tables, authored or reviewed drafts of the paper, participated in the writing or technical editing of the manuscript, and approved the final draft.
- Keita Watanabe and Koichiro Sugimoto conceived and designed the experiments, authored or reviewed drafts of the paper, served as scientific advisor, and approved the final draft.
- Atsuko Ikenouchi and Yukunori Korogi conceived and designed the experiments, authored or reviewed drafts of the paper, participating investigator, and approved the final draft.
- Shingo Kakeda conceived and designed the experiments, authored or reviewed drafts of the paper, participated in the writing or technical editing of the manuscript, and approved the final draft.
- Asuka Katsuki conceived and designed the experiments, authored or reviewed drafts of the paper, participating investigator, provided study patients, and approved the final draft.
- Reiji Yoshimura conceived and designed the experiments, performed the experiments, authored or reviewed drafts of the paper, participating investigator, provided study patients, and approved the final draft.

## Human Ethics

The following information was supplied relating to ethical approvals (i.e., approving body and any reference numbers):

The Local Ethics Committee of the University of Occupational and Environmental Health (UOEH) granted Ethical approval to carry out the study (Ethical Application Ref: 第セH25-13号).

## Data Availability

The raw data and code files are available in the Supplementary Files.

## Supplemental Information

Supplemental information for this article can be found online at http://dx.doi.org/10.7717/peerj.9632#supplemental-information.

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
