# Peer review of "The brain-derived neurotrophic factor Val66Met polymorphism increases segregation of structural correlation networks in healthy adult brains"

_PeerJ, doi:10.7717/peerj.9632_

## Round 0.1 · original submission · Major Revisions

Both of the reviewers raise major concerns.

Reviewer 1 ·

Basic reporting

N/A

Experimental design

N/A

Validity of the findings

N/A

Additional comments

In this study, the authors created the brain-structural correlation network on each of two groups of samples with different BDNF val66met genotype followed by comparing their network-level patterns. As a result, they identified the average local clustering coefficient of the BDNF Met carrier's network was significantly higher than the Val homo group, and some other network measurements didn't show a significant difference.

Overall this is an interesting work. The writing is clear and organized. However, some major considerations need to be taken into account.

Major:
1. I am very confused about what's the biological meaning underlying such a structure co-variance network. Moreover, if there are two brain regions, and the correlation coefficient of their structural features is different between the two groups, then what's the represented potential biological meaning? It would be more straightforward and interesting to directly compare the structure difference in the individual region between the two genotype groups.

2. The authors treated the positive and negative correlations equally using absolute value. Back to my first question, if we are not sure the biological meaning of such a correlation between two regions, it's hard to figure out what's a better way to treat the sign of the correlation.

3. A little curious that is there any difference between Met/Val and Met/Met groups? It seems that currently, the authors treated them as the same group, by which the variance between them was ignored.

4. The sample sizes of the two groups are way more different (16 vs. 33). Some of the comparisons seemed not really fair, such as "a grid pattern was more clearly perceived in the Met carrier heat map than in the Val homozygote heat map."

5. A static power analysis or discussion was necessary, otherwise, the correlations, as well as the comparison on that, were not convinced me to the conclusions.

6. The authors failed to provide clear biological interpretations of their computational results. For example, they identified " Val homozygote networks consisted of weak connections whereas the Met carrier networks had more connections with middle strength.", then, what can we learn from this finding to explore the Met mutation influenced brain function?

7. The most important issue of the paper is that all the findings were generated from a small data set without any validations using other independent resources. Meanwhile, the data has a high variance on gender and age. There is limited information on other factors that could potentially influence the comparison. Consequently, from which the conclusions raised were not fully convinced me.

Minor:
1. The font of some figures is too small.

2. For figure.2, it's really confused to include positive and negative correlations in the same plot. It's challenging to understand what the heatmap wanted to introduce.

3. For figure 3, I didn't find any obvious difference between those two plots.

·

Basic reporting

no comment

Experimental design

no comment

Validity of the findings

no comment

Additional comments

The authors here present structural covariance networks derived from inter-regional cortical thickness correlations, specifically focusing on differences between BDNF
valine (Val) homozygotes and methionine (Met) carriers. The authors found Met carriers’ structural covariance networks have denser connections with neighbors and are more distant from random networks than Val homozygotes’ networks.

The author has made an interesting research, but the significance of conducting this research should more clear.

1. The English language should be improved to ensure that an international audience can clearly understand your text. Some sentences are hard to understand. A careful proofread would be beneficial.

2. Abstract:
Line 46: Should the fourth part of the abstract be a ‘Conclusion’ or a ‘Discussion’? Please check the abstract again.
Line 50: ‘This could be the result of deficits in the BDNF Met carriers in axonal maintenance or synaptic revision’. This sentence is inappropriate here, the article did not conduct further research on this. Delete it would be better.
3. Introduction:
In the third paragraph of this part, the author lists many examples, which should be summarized to show the relationship with the current study would be better.
Line 94-95: Please provide relevant references to support ‘…. brain network will be important for understanding the neural basis of human brain function and disease…..’
Line 105-107: ‘If structure……brain structural networks’, the expression of this sentence is too absolute.
The introduction should be more concise.
4. The title ‘Methods’ should be ‘Materials & Methods’ according to the Standard Sections of ‘Peer J’.
Line 146: A single sentence into a paragraph is not good.
In the section of ‘MRI Acquisition’: Even all the subjects are the doctors with no health issues. Only one sequence, 3D-FSPGR, is collected, and it is not sensitive to the find of brain structure abnormal. I want to know whether there are any other sequences were acquired, such as T2WI and FLAIR, it can sensitively find brain lesions, which is more helpful for the subjects’ inclusion. You know, structure changes in the brain can have a potential impact on the results of the study. Furthermore, the detail of exclusion criteria do not mention in the study.
Line 183: what’s the mean for ‘….0.47kakeru×0.47×…..’
Line 222: double ‘(Váša et al., 2018)’ ?
Line 228: ‘using 31 subjects’ brains included in the Met carrier group and the 16 subjects’ brains included in the Val homozygote group respectively’, so, what the other 2 subjects ? (49 participants).
Line 310: the number of nonparametric permutation test repetitions was 2000, why not 5000 or 1000? Please give an explanation or give the supporting literature.
5. Results: Line 355: ‘p-value ≥ 0.05’ , but in the abstract was ‘p values ≧ 0.05’, The expression of symbols should be unified, please check the whole text.

It would be better for the more concise section of post-processing methods and results

6. Discussion: More in-depth discussion regarding to current findings would be better.

7. Conclusion:
It would be better to delete the sentence ‘This might be the result of deficits in axonal maintenance or synaptic revision.’ There is no relevant research in the current study. Rewrite the sentence ‘Understanding the effect of the BDNF Val66Met SNP on brain structure will help us to understand human intelligence and disease.’ It is too absolute.

Overall, the authors present interesting results with Met carriers’ structural covariance networks have denser connections with neighbors and are more distant from random networks than Val homozygotes’ networks. However, the authors need to better explain the significance of conducting this research.

---

## Round 0.2 · accepted · Accept

All the reviewers have accepted the manuscript.

Reviewer 1 ·

Basic reporting

no comment

Experimental design

no comment

Validity of the findings

no comment

Additional comments

The authors have satisfactorily responded to all my questions and made the necessary changes to the manuscript. I have no further comments.